# Enhancing fraud detection in the Ethereum blockchain using ensemble learning

Zhexian Gu[1,2,3] and Omar Dib[1,2,3]

[1] Department of Computer Science, Kean University, Union, New Jersey, United States
[2] Computer Science and Artificial Intelligence Center, Wenzhou Kean University, Wenzhou, Zhejiang, China
[3] Department of Computer Science, Wenzhou-Kean University, Wenzhou, Zhejiang, China



## ABSTRACT

The Ethereum blockchain operates as a decentralized platform, utilizing blockchain technology to distribute smart contracts across a global network. It enables currency and digital value exchange without centralized control. However, the exponential growth of online commerce has created a fertile ground for a surge in fraudulent activities such as money laundering and phishing, thereby exacerbating significant security vulnerabilities. To combat this, our article introduces an ensemble learning approach to accurately detect fraudulent Ethereum blockchain transactions. Our goal is to integrate a decision-making tool into the decentralized validation process of Ethereum, allowing blockchain miners to identify and flag fraudulent transactions. Additionally, our system can assist governmental organizations in overseeing the blockchain network and identifying fraudulent activities. Our framework incorporates various data pre-processing techniques and evaluates multiple machine learning algorithms, including logistic regression, Isolation Forest, support vector machine, Random Forest, XGBoost, and recurrent neural network. These models are fine-tuned using grid search to enhance their performance. The proposed approach utilizes an ensemble of three distinct models (Random Forest, extreme gradient boosting (XGBoost), and support vector machine) to further improve classification performance. It achieves high scores of over 98% across key classification metrics like accuracy, precision, recall, and F1-score. Moreover, the approach is suitable for real-world usage, with an inference time of 0.13 s.

## INTRODUCTION

Cryptocurrencies like Ethereum (*Buterin, 2013*; *Kushwaha et al., 2022b*) and Bitcoin (*Nakamoto, 2008*; *John, O'Hara & Saleh, 2022*) have experienced a remarkable increase in demand since their inception. These decentralized digital currencies are designed to empower individual users by shifting control away from centralized authorities (*Raskin & Yermack, 2018*). By leveraging blockchain technology, these cryptocurrencies ensure transparency, security, and immutability of transactions, eliminating the need for traditional intermediaries such as banks and financial institutions (*Dib et al., 2018*). This decentralization not only enhances the privacy and autonomy of users but also reduces transaction costs and processing times (*Singh & Kim, 2019*). As a result, cryptocurrencies

Corresponding author
Omar Dib, odib@kean.edu

have gained significant traction as viable alternatives to conventional financial systems, attracting substantial interest and investment from diverse sectors, including finance, technology, and retail. This growing adoption underscores the transformative potential of blockchain technology in reshaping the future of digital transactions and data management (*Dib, Huyart & Toumi, 2020*). The key benefits of cryptocurrencies in modern finance are illustrated in Fig. 1.

Ethereum is a prominent example of a cryptocurrency system, offering cost-effective, user-friendly digital transactions with minimal fees and global access (*Laurent, Brotcorne & Fortz, 2022*). Like other cryptocurrencies, Ethereum operates on a blockchain network, serving as a decentralized and distributed public ledger responsible for verifying and recording transactions. In addition to cryptocurrency transactions, Ethereum enables the use of smart contracts, which are self-executing contracts with the terms of the agreement directly written into code (*Kushwaha et al., 2022a*). This functionality expands the potential applications of Ethereum beyond simple transfers, making it a versatile and powerful tool in the digital economy.

Ethereum's inherent confidentiality provides organizations and institutions with a shield from direct accountability for blockchain activities (*Chen et al., 2020*). However, this veil of anonymity poses significant challenges in identifying wrongdoers in fraud cases, making it difficult to pinpoint individual responsibility. Consequently, the combination of organizational anonymity and user privacy increases the potential for fraudulent behavior within the Ethereum ecosystem (*Zhou et al., 2022*). While this anonymity protects users' privacy and promotes decentralization, it inadvertently creates an environment where malicious actors can exploit the system without fear of repercussions. Therefore, the Ethereum network must balance the benefits of decentralization and confidentiality with robust mechanisms to detect and deter fraudulent activities, ensuring the integrity and trustworthiness of the blockchain. A list of the different security issues in the Ethereum blockchain is depicted in Fig. 2.

The paradigm shift towards this new digitized financial transaction system has, therefore, led to an unprecedented surge in fraudulent activities, presenting substantial challenges for financial institutions and consumers (*Ryman-Tubb, Krause & Garn, 2018*). This rise in fraud and identity theft has become increasingly prevalent, underscoring the urgent need for robust detection and prevention measures (*Chaquet-Ulldemolins et al., 2022*). As the financial landscape evolves, traditional security protocols are often insufficient to address the sophisticated techniques fraudsters employ. Consequently, the need for advanced solutions like artificial intelligence and machine learning to identify and mitigate fraud in real time has become critical (*Dib, Nan & Liu, 2024*). Integrating these technologies is essential to enhancing security, protecting consumers, and maintaining trust in the digital economy.

Traditional fraud detection methods, while somewhat effective, struggle to keep up with fraudsters' evolving tactics. On Ethereum, these challenges are further compounded by the unique characteristics of Ethereum fraud compared to traditional cyber fraud. Unlike conventional cyber frauds, Ethereum frauds target the Ether token, utilize public keys for user identification, and operate within the publicly accessible transaction records of

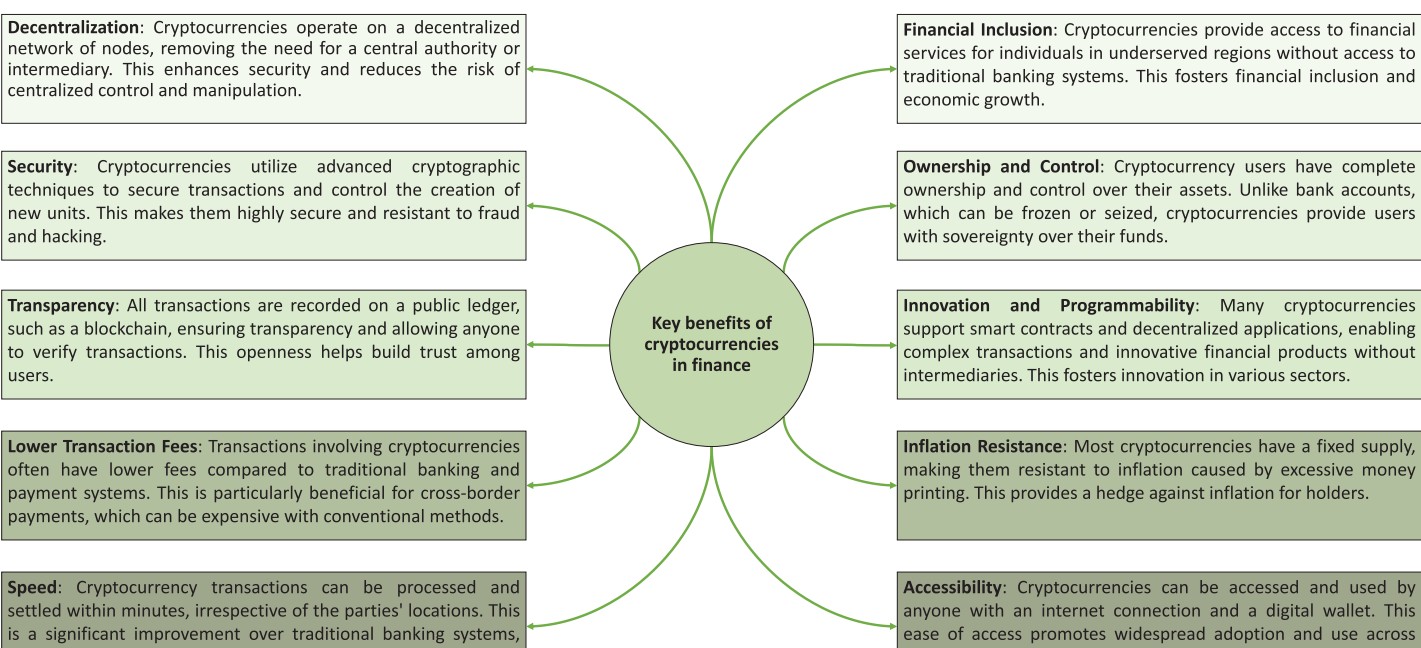

**Figure 1** Core advantages of cryptocurrencies in modern finance.

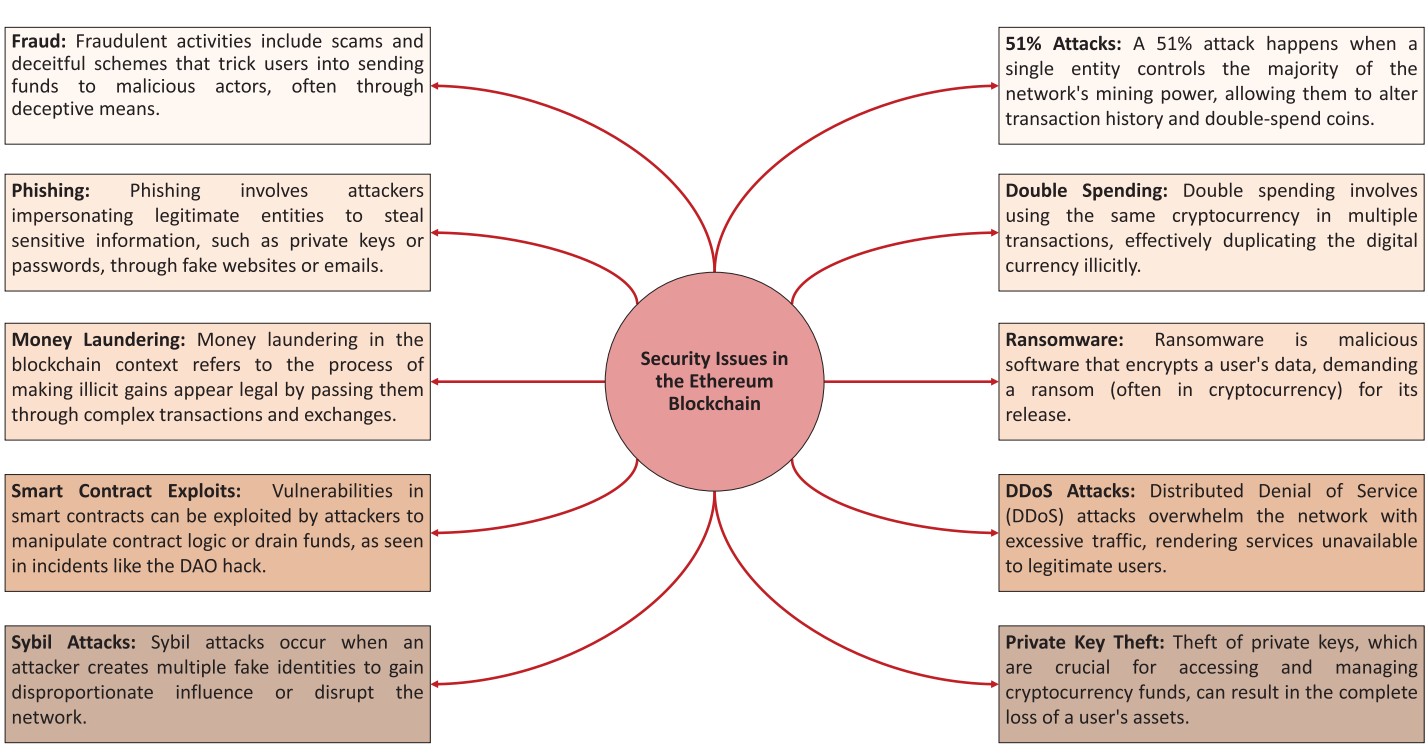

**Figure 2** Overview of key security issues in the Ethereum blockchain.

Ethereum and other digital currency systems (*Tan et al., 2021*). Moreover, the data imbalance issue, where there are significantly fewer labeled fraudulent addresses than legal addresses, presents a significant hurdle in automatically identifying fraudulent transactions within the Ethereum network. In response to these challenges, the field of data science emerges as a crucial ally. Its versatility, effectiveness, and capacity to extract complex patterns—whether normal or fraudulent—from datasets make it invaluable. In fact, data science offers many advanced, robust, and effective techniques and algorithms to detect anomalous behavior indicative of fraudulent activities (*Da'u & Salim, 2020*).

Recognizing the critical need for robust fraud detection tools within the Ethereum blockchain network, this study thoroughly explores the realm of fraud detection by leveraging a diverse range of machine learning and anomaly detection methodologies. This work presents a comprehensive tool that utilizes advanced algorithms and data analytics to significantly enhance the security and integrity of the Ethereum blockchain, thereby ensuring a safer and more reliable environment for digital transactions.

### Contributions

The main contributions of this article are outlined as follows:

- Discussion of the utilization of a decision-making tool based on machine learning and learning data within the Ethereum blockchain framework.
- Proposal of a decision-making tool for identifying fraudulent transactions within the Ethereum blockchain, employing various machine learning techniques and algorithms.
- Utilization of ensemble learning as a collaborative classifier to effectively distinguish fraudulent transactions from normal transactions with high confidence.
- Provision of a comprehensive experimental analysis focusing on various classification algorithms and ensemble learning techniques.

### Article structure

The remainder of the article is organized as follows: "Related Work" reviews existing literature on applying machine learning to detect fraudulent transactions in various sectors. "A Fraud Detection System Within the Ethereum Network" highlights the significance of fraud detection tools as a decision-making aid within the Ethereum ecosystem. "Proposed Framework" elaborates on the dataset and proposes a classification framework. "Experimental Study" details the experimental setup and presents results highlighting classification metrics across various algorithms. This section also includes a comparative analysis against existing fraud detection models. Finally, "Conclusion" summarizes the article's contributions and explores potential directions for future research and improvement.

## RELATED WORK

Fraud detection has been a critical concern in various industries, particularly finance, where the consequences of fraudulent activities can be substantial (*Ryman-Tubb, Krause & Garn, 2018*). Traditional fraud detection methods have relied on rule-based systems (*Engels, Kumar & Philip, 2021*). However, these methods often fall short in adapting to the

evolving nature of fraud and detecting sophisticated schemes effectively (*Öztürk & Usul, 2020*). Consequently, there is increasing interest in utilizing machine learning (ML) and deep learning (DL) techniques to enhance traditional methods and improve fraud detection accuracy. This section reviews the literature on fraud detection methodologies, focusing on ML techniques, anomaly detection methods, DL approaches, hybrid models, evaluation metrics, and benchmark datasets. The aim is to identify the strengths and limitations of current methods and highlight potential research directions.

## Rule-based system

A rule-based system is an expert system that employs a set of "if-then" rules to apply domain-specific knowledge and heuristics for solving problems within a specific and practical problem domain (*Öztürk & Usul, 2020*). Rule-based expert systems offer advantages such as applicability, ease of implementation, and understandability. However, they are constrained by the need for manual rule creation and maintenance, limited capability to manage complex or uncertain scenarios, and susceptibility to inaccuracies if rules are not regularly updated (*Öztürk & Usul, 2020*). While rule-based systems have distinct advantages and limitations, recent literature highlights a trend toward combining them with artificial intelligence to enhance their effectiveness (*Hilal, Gadsden & Yawney, 2022*; *Güneysu, 2023*; *Bellomarini, Laurenza & Sallinger, 2020*).

## AI-based systems

Artificial intelligence (AI) based systems utilize advanced ML algorithms and DL techniques to detect and mitigate fraudulent activities. Unlike rule-based systems, AI-based systems autonomously learn patterns and anomalies from data, allowing them to adapt to evolving fraud schemes. These systems excel in handling complex and uncertain situations by analyzing large datasets and identifying subtle patterns that indicate fraudulent behavior. Moreover, AI-based systems can continuously learn and update their models based on new data, reducing the need for manual intervention and rule maintenance. These models have demonstrated remarkable performance across various sectors. For example, *Dou et al. (2021)* proposed an efficient automated ML approach for predicting the risk of progression to active tuberculosis based on its association with host genetic variations. *Liu et al. (2021)* effectively utilized ML approaches to investigate the relationship between genetic factors and autism spectrum disorder. *Zhenghan & Dib (2022)* applied several ML models to examine the impact of the agriculture sector on the Chinese gross domestic product (GDP). *Chen et al. (2023)* reviewed the application of several ML algorithms for breast cancer diagnosis. These studies highlight the potential of ML algorithms to enhance decision-making processes across various fields.

Recent literature emphasizes the increasing integration of AI into fraud detection systems, with studies showcasing the enhanced detection accuracy and efficiency of AI-driven approaches. The following sections will review the application of various AI-based models for fraud detection.

### Logistic regression

In *Megdad, Abu-Naser & Abu-Nasser (2022)*, logistic regression (LR) achieved an accuracy of 90.0% when operating on an unbalanced dataset. However, LR may underperform in recall and F1-score, particularly with unbalanced datasets. In contrast, *Hamal & Senvar (2021)* found that LR outperforms other algorithms in overall accuracy for detecting fraudulent financial reporting.

### Support vector machine

Many articles, such as *Sivaram et al. (2020)*, *Zhang, Bhandari & Black, 2020*, and *Trivedi et al. (2020)*, state that an support vector machine (SVM) can handle high-dimensional data efficiently, making it suitable for complex datasets. However, SVM is unsuitable for massive datasets due to computational overhead. Additionally, SVM tends to underperform when the feature count of every data point exceeds the number of training data samples.

### Isolation forest

*Zade (2024)* reported that the Isolation Forest (IF) model exhibits poor performance in terms of accuracy, precision, and F1-score both before and after hyperparameter tuning. Its underperformance may be attributed to its sensitivity to parameter settings or the data's lack of distinct anomalous patterns. While the IF algorithm showed moderate recall, indicating some capability to identify fraud, its high rate of false positives poses a significant drawback for practical applications where precision is crucial. Therefore, without substantial model modifications or complementary techniques to enhance its specificity, the IF may not be the most reliable standalone model for detecting fraudulent transactions. *Singh et al. (2024)* stated that one of the advantages of the IF algorithm is its minimal memory allocation requirement and low computational overhead, thanks to its linear time complexity. However, a potential disadvantage of the IF algorithm is that it may struggle with highly imbalanced datasets, which can impact its performance.

### Local outlier factor

*Singh et al. (2024)* stated that the local outlier factor (LOF) identifies outliers by comparing a data point's local variances to those of its neighbors, assessing outliers based on local density influenced by proximity to nearest neighbors. LOF measures the extent of density deviation of a data point from its neighbors, enabling the detection of regions with notably lower densities and emphasizing local relationships and densities to unveil nuanced anomalies often overlooked by traditional methods. This approach enriches anomaly detection granularity, providing a more comprehensive understanding of irregularities within the dataset. One advantage of the LOF algorithm is its ability to capture subtle anomalies that may be missed by other methods due to its focus on local density relationships. However, a potential disadvantage of LOF is its sensitivity to the choice of parameters and the need for careful tuning to achieve optimal performance.

## Random forest

In *Dileep, Navaneeth & Abhishek (2021)*, the authors indicated that the Random Forest (RF) algorithm offers several advantages for financial fraud detection. Firstly, it is an improved version of the decision tree algorithm, combining decision trees to provide better results. RF effectively handles large volumes of decision trees during training and inference, which is the mode of the modules that is beneficial for classification tasks. Additionally, RF is less prone to overfitting than individual decision trees, enhancing the model's generalization capability. However, some disadvantages are associated with RF in financial fraud detection. One potential drawback is the complexity of interpreting the results due to the ensemble nature of the model, which can make it challenging to understand the decision-making process. Another limitation is the computational resources required for training and maintaining a large number of decision trees, which can lead to increased processing time and resource consumption.

## Extreme gradient boosting

*Hajek, Abedin & Sivarajah (2023)* stated that the advantages of extreme gradient boosting (XGBoost) in financial fraud detection include its computational efficiency, scalability, ability to build incremental models to improve predictive power, and ability to minimize overall errors by reducing errors with incremental improvements. XGBoost also improves robustness to noise and overfitting by introducing random sampling schemes with a more normalized model to control overfitting. In addition, XGBoost has advantages when dealing with high-dimensional data and has been successfully applied in areas such as insurance fraud detection. One of the drawbacks of XGBoost is that applying the XGBoost algorithm in the improved XGBOD feature space results in a long execution time, averaging 4,256.25 s. In addition, XGBOD's disadvantages include its long execution time. Still, compared to other anomaly detection methods, XGBOD takes advantage of the labels assigned to mobile transactions and performs well in terms of accuracy and recall rates.

## Recurrent neural networks

In *Nama & Obaid (2024)*, recurrent neural networks (RNNs) excel in modeling sequential data, such as transaction histories, by capturing temporal associations and patterns in financial transactions. They can identify abnormalities in transaction amounts, frequencies, or other relevant factors to detect potentially fraudulent transactions, leveraging the temporal dynamics of the data. RNNs have the capability to evaluate streaming data in real-time, enabling quick responses for fraud detection and loss prevention. Despite their effectiveness in managing unbalanced datasets with few fraudulent transactions, RNNs are often combined with other methods and algorithms, such as ensemble approaches and feature engineering, to enhance overall fraud detection system efficacy and robustness.

Building on the extensive body of research and diverse methodologies explored in the literature, our work aims to address the persistent and evolving challenges in fraud detection within the Ethereum blockchain network. By integrating state-of-the-art ML and anomaly detection techniques, we leverage the Ethereum Fraud Detection Dataset initially

referenced by *Farrugia, Ellul & Azzopardi (2020)*. This dataset serves as a foundation for testing the effectiveness of our approach, providing real-world transaction data to train and validate models capable of identifying fraudulent activity. While traditional rule-based systems have been instrumental in early fraud detection efforts, their limitations in adaptability and handling complex, evolving fraud patterns have paved the way for AI-based systems that leverage advanced algorithms and continuous learning.

Our approach builds on this shift toward AI by combining a variety of cutting-edge techniques. We employ sophisticated data preprocessing methods that enhance the quality and relevance of input data, ensuring that the models can focus on the most important features. Hyperparameter tuning is incorporated to optimize model performance, as even slight adjustments can yield significant improvements in fraud detection accuracy. Furthermore, we integrate multiple ensemble learning methods—such as voting, stacking, and boosting classifiers—which allow us to combine the strengths of different models, reducing biases and improving overall predictive power. These ensemble methods also increase robustness, as they can mitigate overfitting and improve the model's ability to generalize across different datasets and environments. By positioning our work within the context of these advancements, we demonstrate how our model matches and exceeds the performance benchmarks set by previous studies, offering significant improvements in precision, recall, and computational efficiency. This research, therefore, contributes significantly to the ongoing evolution of fraud detection technologies in the digital world, particularly in the realm of e-commerce and other online transactions. By providing a robust and scalable solution, our work enhances the security and integrity of digital transactions on the Ethereum blockchain, ensuring a safer environment for users and stakeholders in the rapidly growing digital economy.

## A FRAUD DETECTION SYSTEM WITHIN THE ETHEREUM NETWORK

In this section, we elaborate on how the proposed fraud detection system is seamlessly integrated into the decision-making process within the Ethereum consensus mechanism. Additionally, we provide a detailed explanation of its practical applicability by illustrating a real-world scenario, demonstrating how the system enhances transaction validation and strengthens network security.

The proposed approach for detecting fraudulent transactions on the Ethereum blockchain begins with the initiation of a transaction. A native transaction object is constructed, including critical parameters such as Nonce, Gas Price, Gas Limit, destination address, sending amount, and transaction data. The sender signs the transaction using their private key to confirm ownership and authorization. This signed transaction is then submitted to a local node for initial verification of legitimacy and authenticity.

At this stage, the proposed fraud detection module, embedded within local nodes and miner nodes, applies ML algorithms to analyze the transaction for signs of anomalous or malicious behavior. These algorithms thoroughly examine patterns in the transaction's metadata, such as unusual gas prices, transaction sizes, or the frequency of transactions originating from specific addresses. Transactions flagged as suspicious are temporarily

isolated and subjected to further scrutiny, while legitimate transactions proceed to the next phase.

Once verified by the ML module, legitimate transactions are added to the transaction pool managed by miner nodes, where they await inclusion in a block. The transaction is then broadcast across the Ethereum network, allowing other nodes to validate and process it. When a miner node successfully solves the proof-of-work problem and discovers a valid block, it includes the transaction alongside others in the block. This block is then broadcast to the network for synchronization with the distributed ledger.

To ensure the security and finality of the transaction, it undergoes multiple block confirmations. During this process, additional instances of the fraud detection module embedded in local nodes continuously monitor the transaction chain for any inconsistencies or anomalies. This multi-layered approach enhances transaction integrity and network security. The framework illustrating the main steps of deploying the fraud detection system within the Ethereum network is presented in Fig. 3.

To further substantiate the practical utility of the proposed system, we consider the case of a decentralized finance (DeFi) platform built on Ethereum. Fraudulent activities, such as flash loan attacks or smart contract exploits, can be detected through the system's ML module, which analyzes transaction patterns in real time. For instance, a series of rapid transactions with unusually high gas prices might indicate a flash loan attack. By flagging such transactions early, the system prevents them from being included in a block, mitigating financial losses and safeguarding platform users. Additionally, consider phishing scams where compromised wallets suddenly generate high volumes of outgoing transactions. The fraud detection module can identify these anomalies based on the wallet's historical behavior, alerting validators to halt suspicious transactions.

## PROPOSED FRAMEWORK

This section outlines the framework for analyzing and detecting fraudulent transactions within the Ethereum network using the Ethereum Fraud Detection Dataset (*Farrugia, Ellul & Azzopardi, 2020*). The key components of the framework are presented in Fig. 4 and include data exploration and preprocessing, data split, feature engineering, model selection, and tuning. This framework integrates various data preprocessing techniques and machine learning models to accurately classify transactions as fraudulent or legitimate.

### Data preprocessing

The following discusses the data preprocessing steps to prepare the dataset (*Farrugia, Ellul & Azzopardi, 2020*) for effective learning. The dataset can be publicly accessed here: *Vagifa (2024)*. Since proper preprocessing is essential for effective training and accurate classification of fraudulent transactions, we applied a range of preprocessing techniques to handle the complexity and diversity of the dataset.

The Ethereum Fraud Detection Dataset initially referenced by *Farrugia, Ellul & Azzopardi (2020)* was primarily loaded with 50 features, each comprising 9,841 samples. After examination, the 'Index' and 'Address' features were deemed irrelevant for subsequent fraud detection and removed. The remaining 48 features were divided into

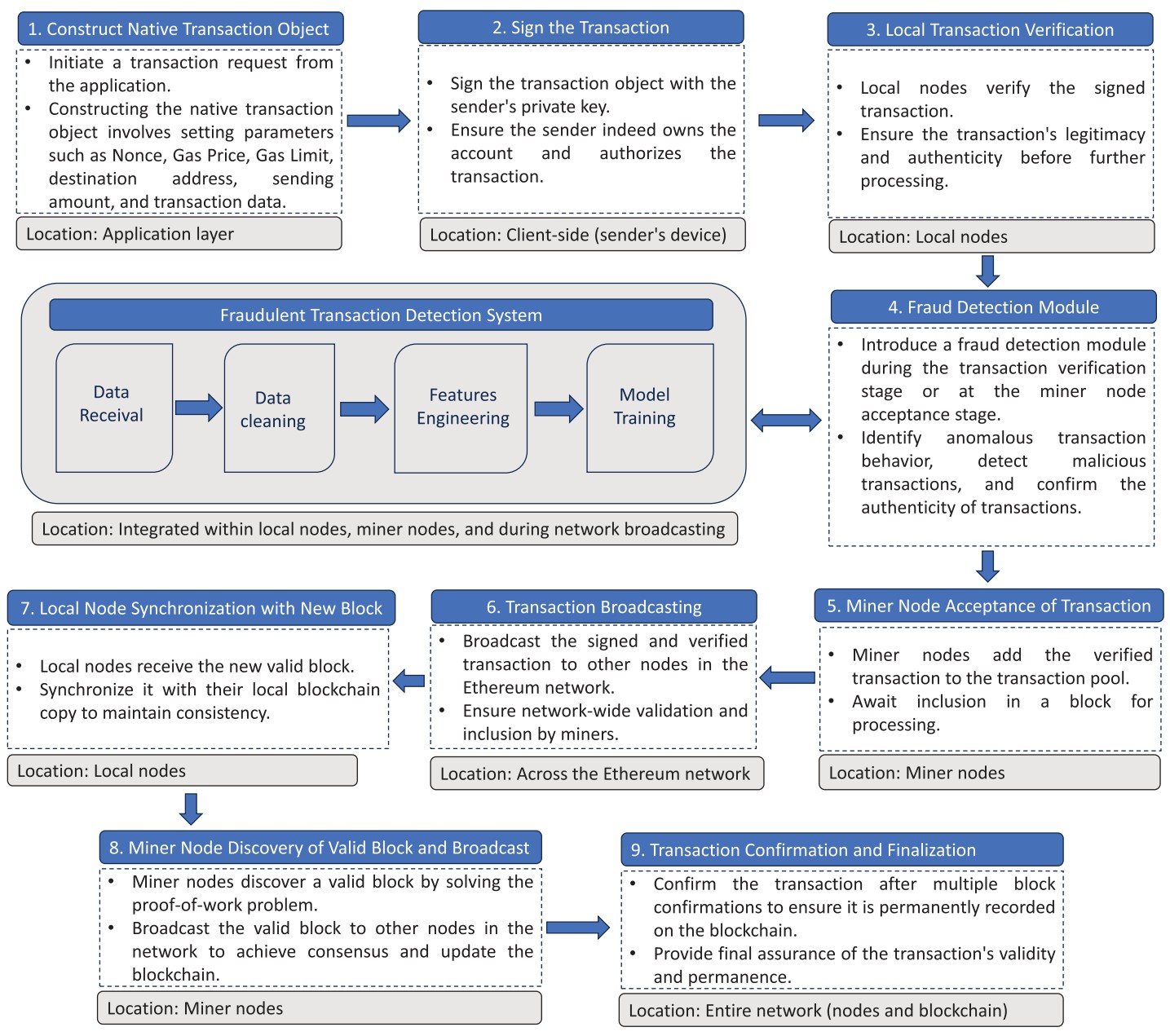

**Figure 3** Integration of a fraud detection system into the Ethereum blockchain infrastructure.

numerical and categorical groups based on their data types: numerical and object. Two of the 48 features were identified as categorical variables with high cardinality. These were excluded from further analysis to ensure more efficient and effective learning. Regarding the numerical features, the initial investigation centered on the 'FLAG' feature, which indicated whether a transaction was fraudulent. A pie chart analysis of this feature revealed an imbalanced dataset, with 77.86% of the data classified as non-fraudulent and 22.14% as fraudulent, as shown in Fig. 5. *Zhang et al. (2024)* stated that selecting the most effective

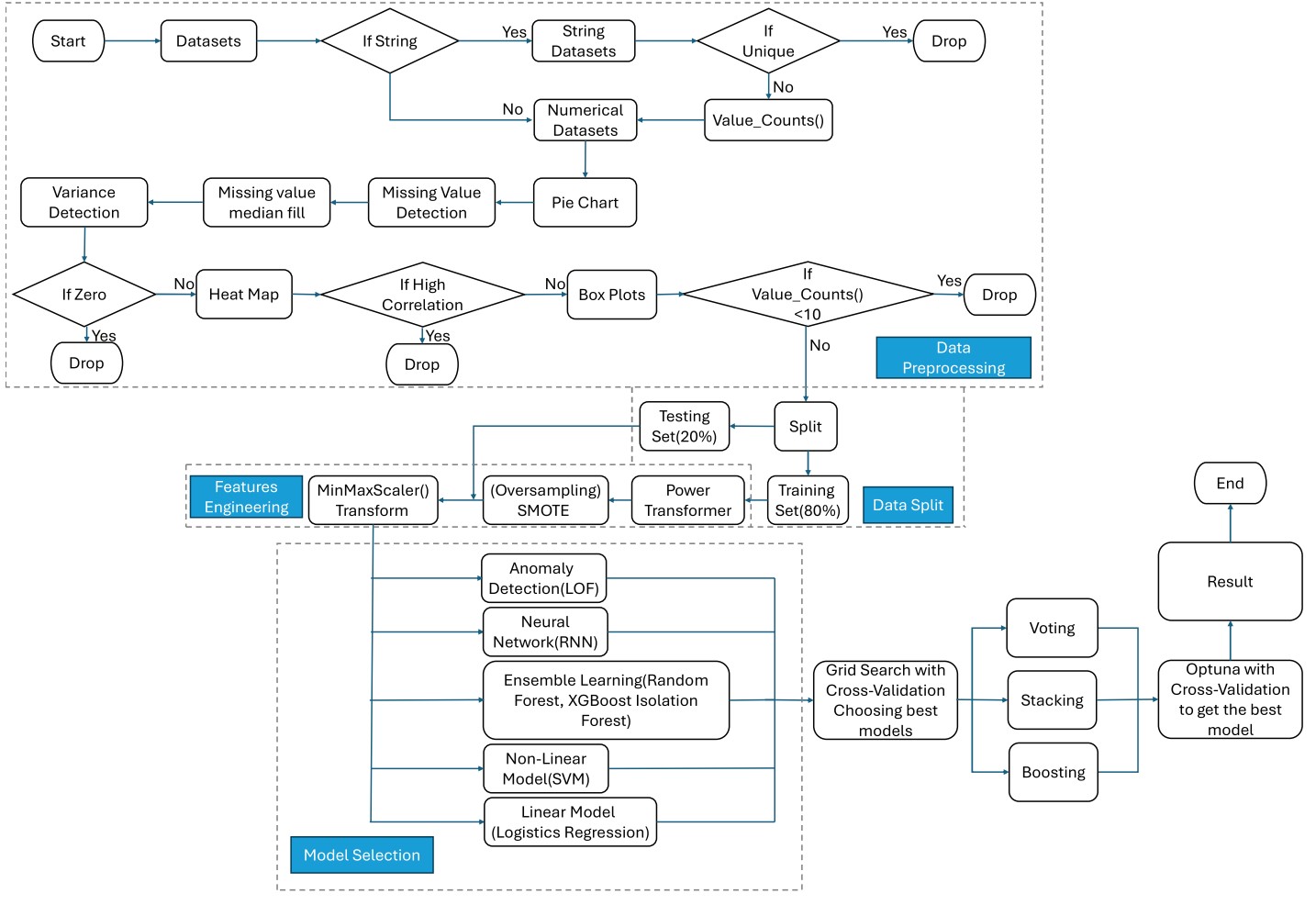

**Figure 4 Architecture for financial fraud detection leveraging ensemble machine learning.**

active learning strategy in imbalanced datasets can be challenging due to the skewed distribution of classes.

Next, missing values were visualized in Fig. 6, and a median imputation method was applied to fill the gaps. Missing data is a common challenge in datasets, often resulting from the absence of values for certain variables in specific observations. As noted by *Gupta et al. (2024)*, missing data can introduce bias and significantly impact the performance and reliability of predictive models, potentially leading to misleading results. To mitigate this, *Lee & Yun (2024)* suggests that median imputation, by reflecting relationships between influential factors, can provide a robust solution compared to other methods. However, the imputation technique, while useful, may not completely address the underlying causes of missing data, such as systemic biases in data collection. Additionally, variance analysis was conducted to identify features with zero variance, which were subsequently dropped due to their lack of significance. Zero variance features are listed below:

Target distribution

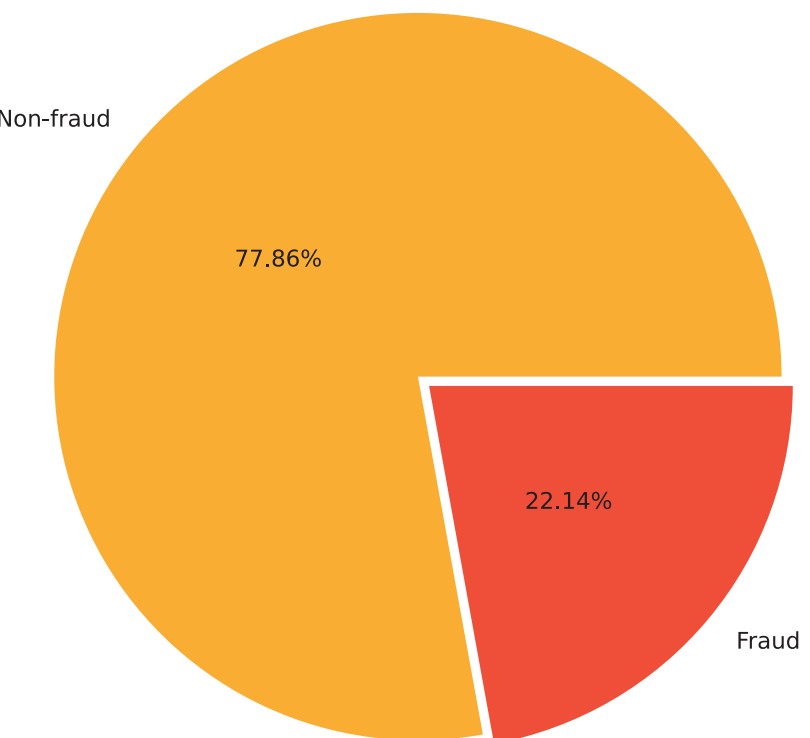

Non-fraud

77.86%

22.14%

Fraud

**Figure 5 Distribution of fraud and non-fraud instances illustrated *via* a pie chart.**

- ERC20 avg time between sent tnx
- ERC20 avg time between rec tnx
- ERC20 avg time between rec 2 tnx
- ERC20 avg time between contract tnx
- ERC20 min val sent contract
- ERC20 max val sent contract
- ERC20 avg val sent contract

A correlation matrix in Fig. 7 identified and removed highly correlated numerical features to enhance training efficiency by reducing redundancy.

By observing the correlation matrix, the following features are dropped:

- Avg value sent to contract
- ERC20 min val sent
- ERC20 max val sent
- ERC20 avg val sent
- Max val sent to contract

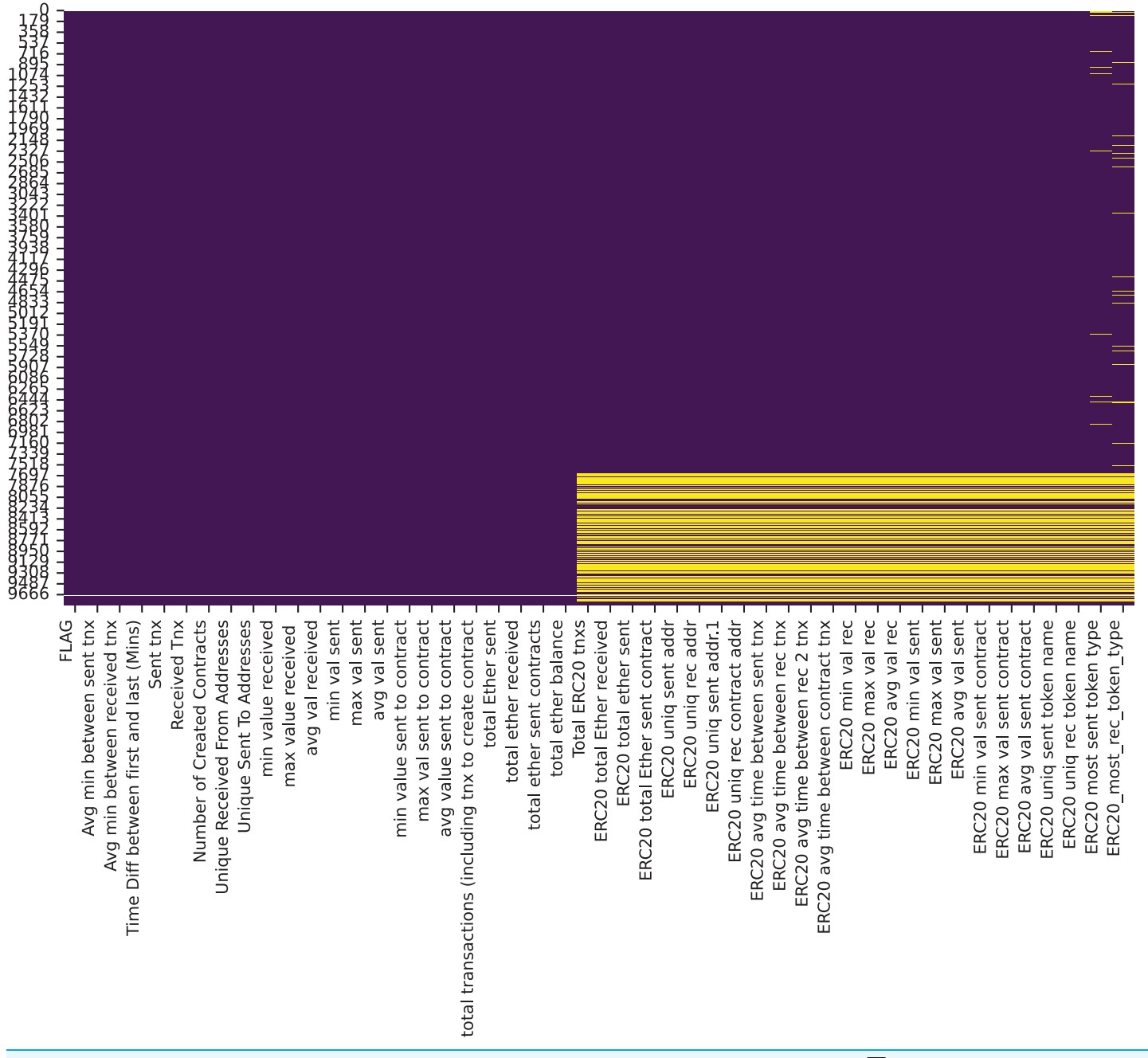

**Figure 6  Visualization of missing values in the dataset.**               

- Total ether sent contracts
- Time Diff between first and last (Mins)
- Total ether balance
- ERC20 max val rec
- ERC20 uniq rec token name

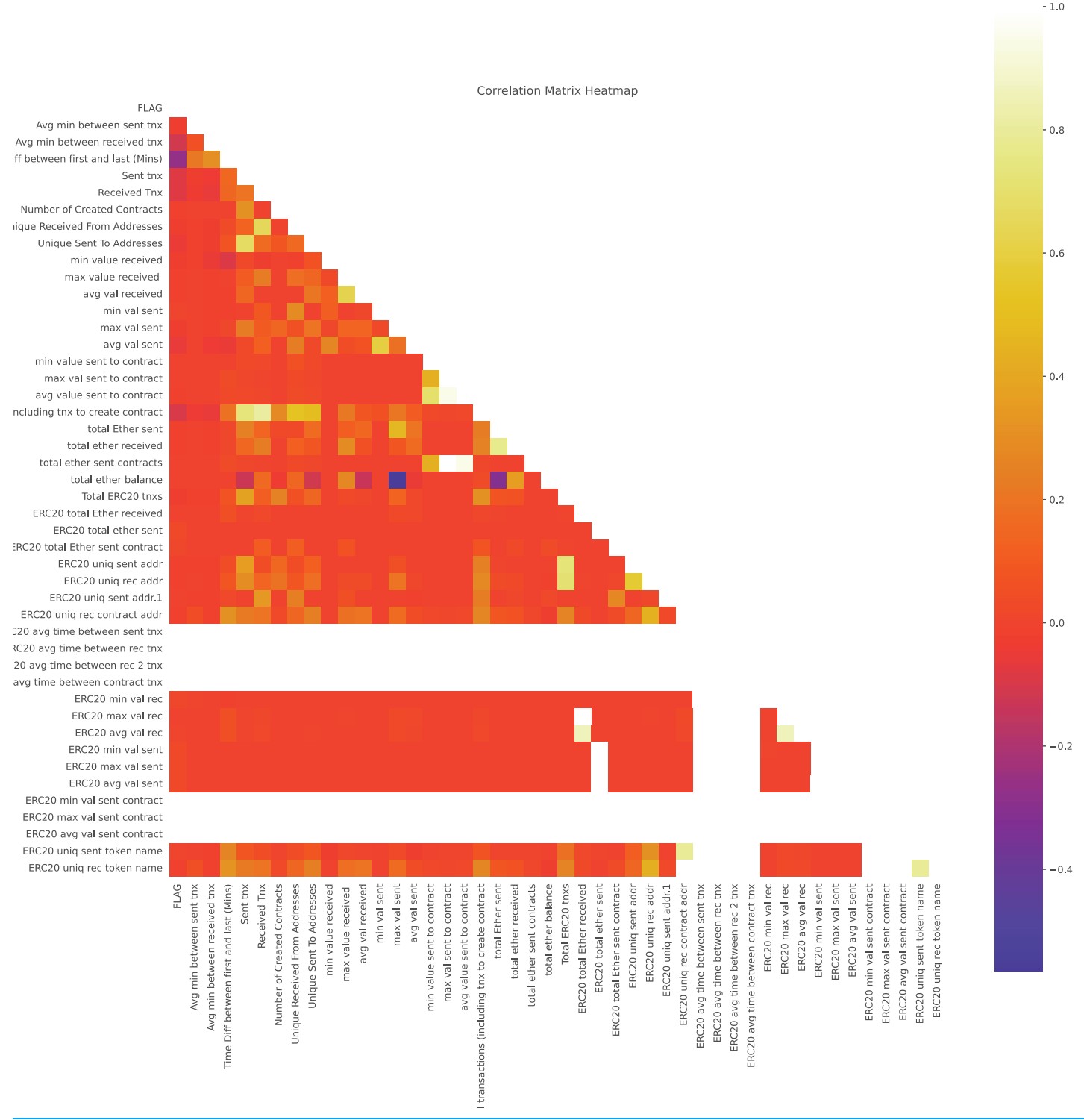

**Figure 7 Heatmap of feature correlations to evaluate variable interdependencies.**

- ERC20 avg val rec
- Total transactions (including tnx to create contract)
- ERC20 uniq sent token name
- Total ERC20 tnxs

After dropping the following features, the correlation matrix becomes Fig. 8

Boxplot analysis, as illustrated in Fig. 9, aids in visualizing the distribution of each feature, facilitating the identification of features exhibiting tightly clustered quartiles or extreme skewness. Anomalies in distribution can potentially distort data analysis outcomes. Subsequent refinement involved utilizing the value_counts method, with a threshold of 10 for unique values. Features with fewer than 10 unique values were deemed insignificant and therefore excluded. Consequently, the following features were dropped:

- ERC20 unique sent addresses
- Minimum value sent to contract

### Data split

The dataset was divided with 20% allocated to the test set and 80% to the training set. This allocation ensures a representative sample for evaluating model performance while providing a larger set for training the model, enabling it to effectively learn patterns and relationships within the data. By reserving a separate portion of the data for testing, the study ensures an unbiased evaluation of the model's accuracy and generalization capabilities on unseen data.

### Feature engineering

The Boxplot observations revealed that the dataset features displayed non-normal distributions. To enhance model performance and bring the features' distributions closer to normal, we implemented a PowerTransformer transformation. This transformation aids in stabilizing the data distribution, mitigating skewness, and facilitating improved model learning. Following the PowerTransformer transformation, boxplots resemble those depicted in Fig. 10.

After examining Fig. 10, it became evident that five features still exhibited poor distribution. Further investigation revealed that many of these features contained many zero values. Efforts were thus made to tackle this issue by employing various imputation methods, such as median and mean filling, but no significant improvement in feature distribution was observed. Additionally, dropping these features entirely and evaluating performance using logistic regression, random forest, and XGBoost classifiers negatively impacted model accuracy.

To address the dataset's imbalance, particularly with respect to fraudulent transactions, we employed the SMOTE (Synthetic Minority Over-sampling Technique) method to enhance the model's ability to detect fraudulent activities, which typically constitute the minority class. SMOTE works by generating synthetic samples for the underrepresented class rather than simply replicating existing data points. This process involves selecting a
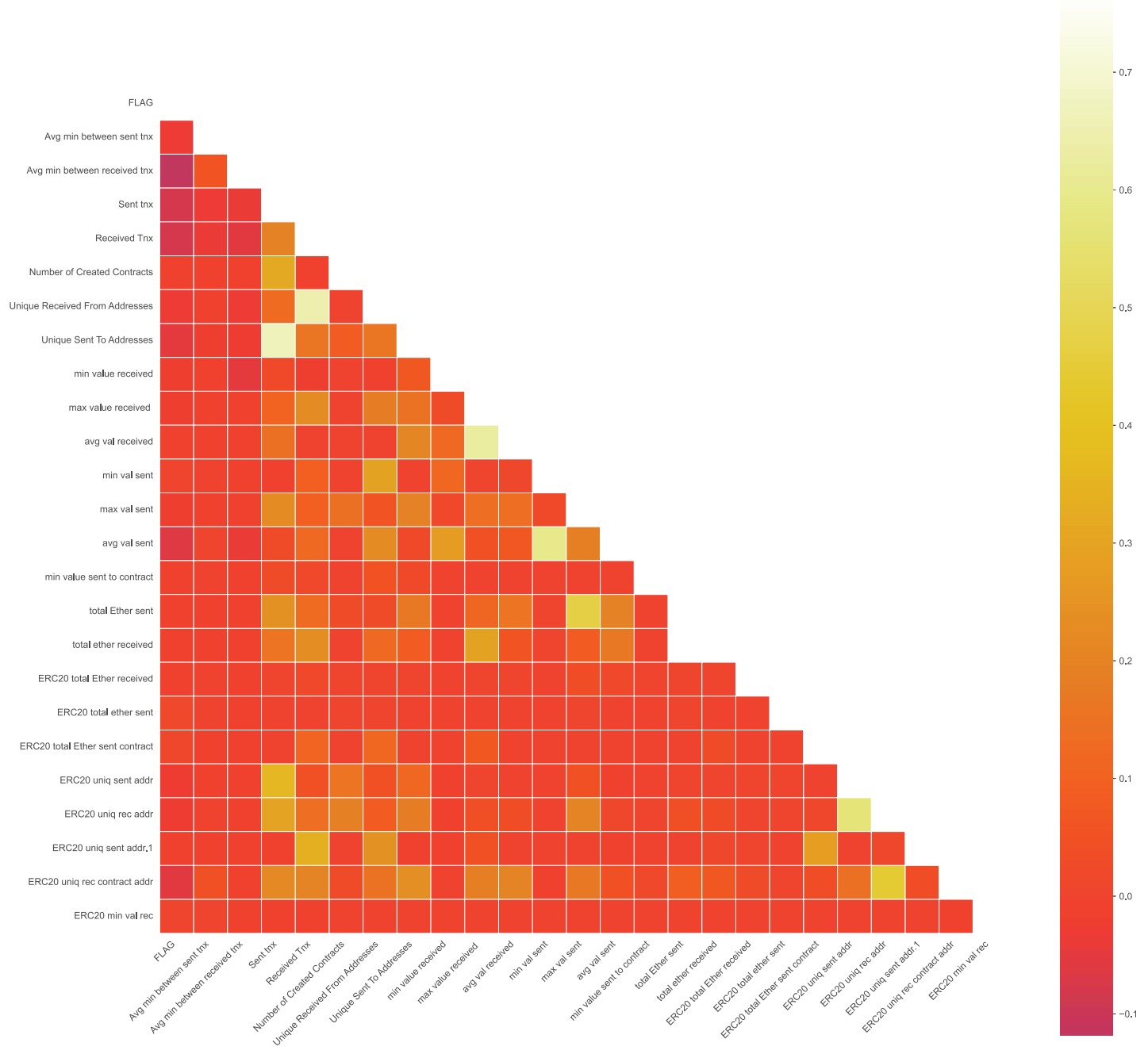

**Figure 8** **Updated correlation matrix after eliminating highly correlated features.**

data point from the minority class and generating new synthetic samples along the line segments joining it to its nearest neighbors. By increasing the number of samples in the minority class, SMOTE helps balance the class distribution, reducing the bias that would otherwise favor the majority class during training. This technique improves the model's sensitivity and recall for detecting fraud, ensuring that the predictive performance is not

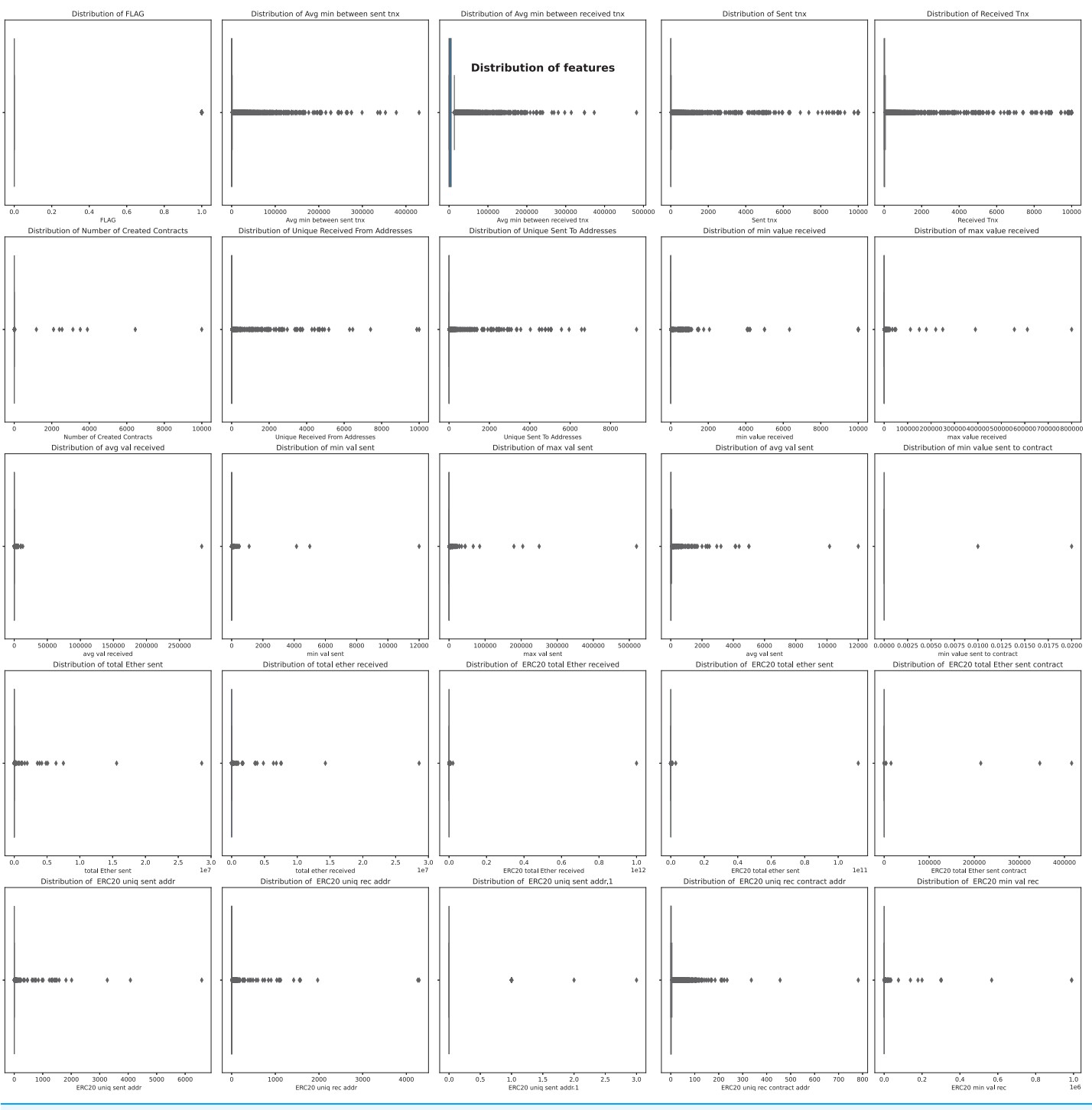

**Figure 9** Boxplot analysis for identifying outliers and distribution patterns.

skewed toward the majority class. Furthermore, applying SMOTE enhances the model's generalization capabilities, ensuring better detection of previously unseen fraudulent patterns. Further analysis of the dataset revealed varying ranges of values across different

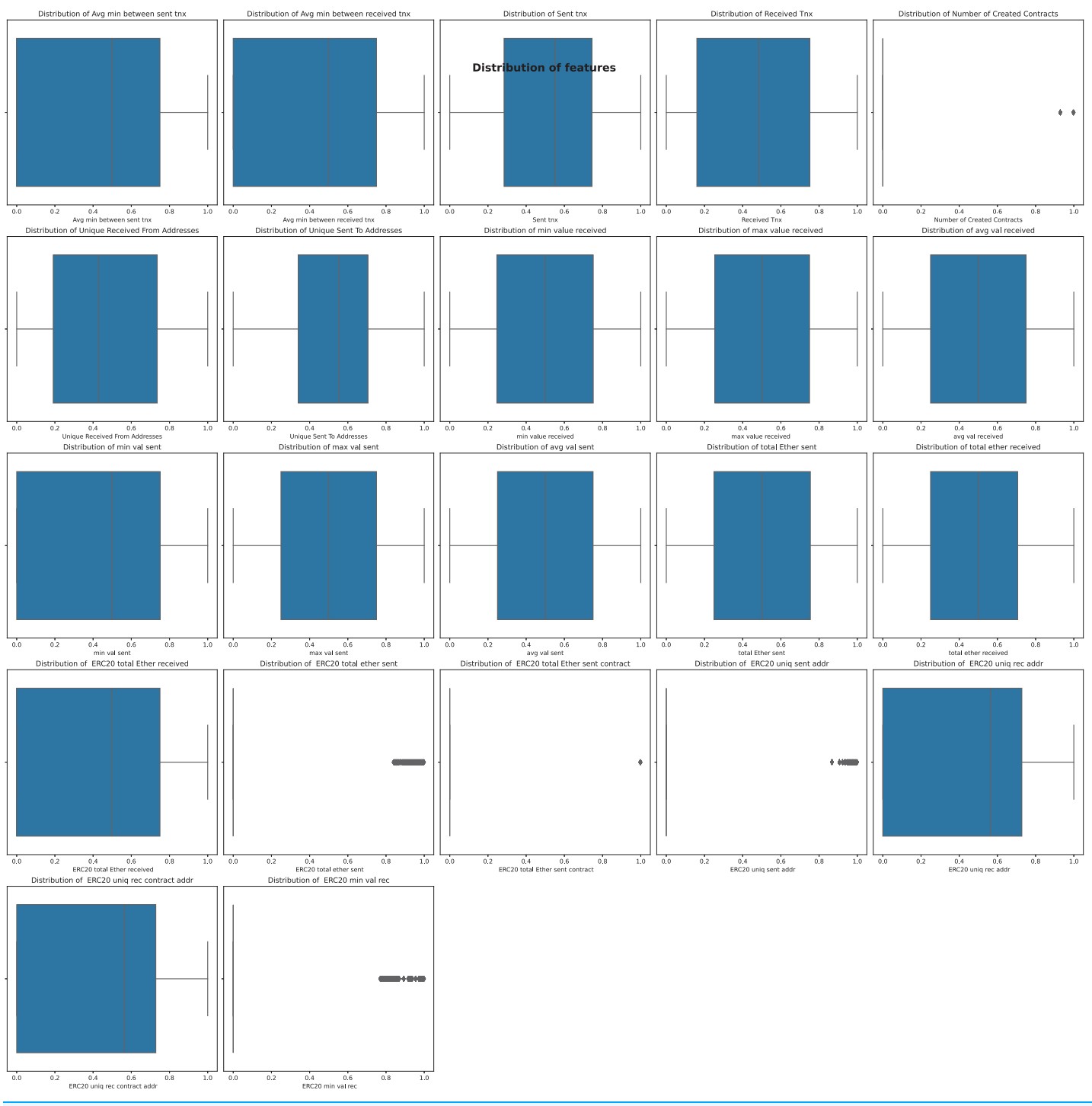

**Figure 10** BoxPlot analysis after applying PowerTransformer.

features. This discrepancy could cause the model to assign disproportionate importance to features with higher numerical values. To address this, we applied MinMaxScaler to normalize the range of feature values, scaling them to a range between 0 and 1. This

normalization eliminates dimensional differences among features and ensures the model treats all features with consistent sensitivity.

## Model training and tuning

After completing data preprocessing, splitting, and feature engineering, the next step involves selecting appropriate machine learning models and training them to detect fraudulent transactions within the Ethereum network. As illustrated in Fig. 4, the model selection phase involves training various machine learning and deep learning algorithms for the classification problem. The selected algorithms include LOF for anomaly detection, RNN, RF, XGBoost, IF, SVM, and LR. Our objective is to evaluate the performance of these different models and analyze their results to achieve optimal classification performance.

The selection of these methods for detecting fraudulent transactions in the Ethereum network was driven by the need for robustness and accuracy in handling complex, high-dimensional data. Each chosen algorithm offers unique advantages suited to different aspects of fraud detection. For example, LOF excels in identifying anomalies in dense data clusters, crucial for spotting irregular transactions. RNNs were leveraged for their ability to capture sequential dependencies, ideal for detecting patterns indicative of fraudulent behavior over time. RF and XGBoost provided robustness against noise and scalability for large datasets, while IF offered efficiency in isolating outliers. SVMs handled non-linear relationships well, and LR provided interpretable results, serving as a reliable baseline. Ensemble methods like voting, stacking, and boosting further consolidated these strengths, enhancing predictive accuracy through model combination and error correction. This comprehensive approach aimed to achieve optimal performance by leveraging the strengths of each method within a cohesive framework tailored to the unique challenges of fraud detection in cryptocurrency transactions.

The following presents the detailed benefits of applying each algorithm for the classification task. LOF is particularly effective at identifying anomalous transactions by measuring the local density deviation of data points. It is suitable for detecting outliers in high-dimensional datasets where fraudulent transactions may appear as anomalies.

RNNs, despite being powerful for sequential data analysis, can also be used for classification tasks. They can capture complex dependencies and detect irregularities in transaction data, making them suitable for spotting fraudulent activities.

RF is robust and versatile, providing high accuracy by combining the predictions of multiple decision trees. It handles large datasets well and can manage various feature types, making it effective for complex fraud detection tasks.

IF isolates observations by randomly selecting features and splitting values, effectively identifying outliers. Its efficiency and scalability suit large datasets with potentially fraudulent transactions.

SVM is effective for binary classification by finding the optimal hyperplane that maximizes the margin between classes. It performs well in high-dimensional spaces and can handle non-linear relationships using kernel functions.

LR is a simple yet powerful linear model for binary classification. It provides probabilities for class membership, making it easy to interpret and implement. It's often a strong baseline model for fraud detection.

After training each algorithm, grid search is used alongside cross-validation to fine-tune the individual models and potentially improve their performance. Grid search systematically explores a range of hyperparameter combinations to identify the optimal settings for each model. Cross-validation, typically k-fold cross-validation, ensures that the model's performance is evaluated robustly by splitting the data into multiple training and validation sets.

To further enhance the classification framework, the top three models in terms of accuracy have been selected as the base for further improvement. These models serve as the base learners for three advanced ensemble learning algorithms: voting, stacking, and boosting.

The voting ensemble method aggregates the predictions of the selected top models by taking a majority vote or averaging their probabilities. This approach capitalizes on the diverse decision-making processes of each individual model, improving overall performance and robustness against overfitting. By combining multiple models, voting could reduce the impact of any one model's weaknesses.

Stacking, on the other hand, involves training a meta-model on the outputs of the base models. Here, the predictions from the top models (RF, XGBoost, and SVM) serve as input features for the meta-model, such as LR or gradient boosting machine. The meta-model learns to correct the errors of the base models, resulting in improved predictive accuracy. This method is particularly useful when the base models complement each other and have different strengths, allowing the stacking model to capture complex patterns that individual models may miss.

Boosting, specifically gradient boosting, sequentially trains models such that each subsequent model focuses on correcting the errors of its predecessor. This iterative process minimizes bias and variance, leading to a more accurate and robust model. Boosting is particularly effective for fraud detection, where detecting subtle and complex patterns is crucial. It helps the model to concentrate on difficult-to-classify fraudulent cases, thus improving detection rates for minority fraud cases, which are typically underrepresented in the dataset.

To further optimize performance, each ensemble method was fine-tuned using Optuna, a cutting-edge hyperparameter optimization framework. Optuna employs advanced techniques such as Bayesian optimization, tree-structured Parzen estimators (TPE), and multi-armed bandit algorithms to efficiently explore the hyperparameter space, identifying the optimal settings for each model. This optimization process ensures that the final models are both accurate and well-calibrated to the specific characteristics of the dataset.

The evaluation of these ensemble methods aims to maximize classification accuracy while enhancing the model's adaptability to evolving fraudulent patterns. By combining multiple models, we aim to improve robustness and reliability, striving to achieve high

detection accuracy and minimize false positives. This approach ensures the system's practical effectiveness in real-world fraud detection applications.

## Performance evaluation

In this subsection, we detail the various techniques employed to rigorously evaluate the performance of machine learning models.

We use both cross-validation and hold-out validation methods to assess the robustness and generalizability of models. The use of these validation approaches helps ensure that the model's performance is not overly reliant on a specific data split and is representative of the entire dataset. Additionally, we use confusion matrices to analyze the model's predictions regarding true positives, false positives, true negatives, and false negatives. From the confusion matrix, we derive key metrics such as accuracy, recall, precision, and F1 score to measure the model's performance on different aspects of classification.

In fraud detection, missing a fraudulent transaction can lead to significant financial losses, making recall a critical metric. Recall ensures the model identifies as many actual fraudulent transactions as possible, thereby minimizing the risk of undetected fraud. Equally important is precision, which measures the model's accuracy in predicting fraud and helps reduce false positives that could otherwise result in unnecessary investigations and wasted resources. Given the need to balance these two metrics, the F1 score is utilized as it provides a harmonic mean of precision and recall, offering a comprehensive assessment of the model's ability to effectively detect fraud while maintaining reliability. These metrics are prioritized to align with the dual goals of minimizing undetected fraud and avoiding excessive false alarms in practical applications.

Accuracy is chosen when the dataset is balanced. Accuracy measures the ratio of the model's overall correct predictions. Accuracy is calculated using the formula:

$$Accurancy = \frac{TP + TN}{TP + TN + FP + FN}. \tag{1}$$

Precision is used to predict how many samples of a positive class in a model are correct. Precision is determined as follows:

$$Precision = \frac{TP}{TP + FP}. \tag{2}$$

Recall allows the model to correctly detect the proportion of the true positive (TP) sample. Recall is computed as follows:

$$Recall = \frac{TP}{TP + FN}. \tag{3}$$

F1 score balances the trade-off between accuracy and recall. F1-score is the harmonic mean of precision and recall:

$$F1 = \frac{2 \times Precision \times Recall}{Precision + Recall}. \tag{4}$$

In addition, since the detection module will operate within the blockchain network, whether on a private or public version of Ethereum, it is crucial to measure the time required for the model to determine whether a transaction is fraudulent or not. Thus, we ensure that the testing time is optimized for real-time performance. Moreover, as the network evolves, new patterns of fraudulent transactions may emerge. Therefore, we have also measured the training time for each algorithm to assess how quickly the models can be retrained with new data, whether using offline or online learning methods. This ensures that the model can adapt swiftly to new fraud patterns, maintaining its effectiveness and reliability.

# EXPERIMENTAL STUDY

## Experimental setup

Experiments and model training were conducted on a Dell Precision 3660 Tower, featuring a 12th Gen Intel Core i9-12900 CPU with 16 cores (2.40 GHz base clock), 64 GB of memory, an NVIDIA RTX A5000 GPU, and running Windows 11. The experimental setup used PyCharm and Anaconda, with Python libraries, including Pandas for data manipulation, Scikit-learn for machine learning and evaluation, Matplotlib for visualization, and XGBoost for gradient boosting. The list of project libraries and requirements can be found in our GitHub repository (*Zhexian & Dib, 2024*).

Based on extensive research on related work, the selected models for the experiments were LR, SVM, IF, LOF, RNN, random forest (RF), and XGBoost. To ensure reproducibility, we have made the complete source code of our project publicly available on GitHub. The repository can be accessed from here *Zhexian & Dib (2024)*.

Initial testing involved running each model with its default parameters to establish a performance baseline. This step aimed to understand the default performance of each model. Subsequently, the impact of hyperparameter tuning on model performance was studied, specifically focusing on how parameter adjustments affect detection in the context of fraud detection. Additionally, the performance of three stacking algorithms based on ensemble learning with the top three individual models was investigated to explore the benefits of ensemble learning in enhancing the classification framework. For all models and experiments, classification metrics for both normal and fraud instances were reported to comprehensively evaluate each model's performance.

In the final part of the study, the best-performing model was fixed and used in a comparative analysis against existing articles and projects utilizing the same dataset, aiming to benchmark the model's performance against the state-of-the-art.

## Experimental results

The performance metric of the original version of different classification algorithms is reported in Table 1 and further illustrated in Fig. 11. The comparative analysis of different models reveals that RF and XGBoost are the most effective for fraud detection, achieving

**Table 1 Performance metrics of different original models.**

| Model | Instance | Accuracy | Precision | Recall | F1-score | Training (s) | Testing (s) |
|---|---|---|---|---|---|---|---|
| RNN | Fraud | 0.86 | 0.64 | 0.82 | 0.72 | 8.88 | 0.50 |
| | Normal | 0.86 | 0.96 | 0.85 | 0.90 | | |
| LR | Fraud | 0.94 | 0.81 | 0.92 | 0.86 | 0.09 | 0.00 |
| | Normal | 0.94 | 0.98 | 0.94 | 0.96 | | |
| LOF | Fraud | 0.73 | 0.22 | 0.10 | 0.14 | 0.18 | 0.10 |
| | Normal | 0.77 | 0.37 | 0.90 | 0.14 | | |
| IF | Fraud | 0.66 | 0.37 | 0.82 | 0.51 | 0.07 | 0.05 |
| | Normal | 0.66 | 0.92 | 0.63 | 0.75 | | |
| SVM | Fraud | 0.57 | 0.26 | 0.55 | 0.35 | 2.21 | 0.25 |
| | Normal | 0.57 | 0.82 | 0.58 | 0.68 | | |
| RF | Fraud | 0.99 | 0.98 | 0.95 | 0.97 | 1.52 | 0.01 |
| | Normal | 0.99 | 0.99 | 1 | 0.99 | | |
| XGBoost | Fraud | 0.99 | 0.97 | 0.97 | 0.97 | 0.32 | 0.00 |
| | Normal | 0.99 | 0.99 | 0.99 | 0.99 | | |

near-perfect accuracy, precision, recall, and F1-scores for both fraud and normal instances, with RF and XGBoost showing accuracy rates of 0.99 for both instance types. RF's precision and recall for fraud detection are 0.98 and 0.95, respectively, while XGBoost achieves 0.97 for both metrics. Logistic Regression (LR) also demonstrates high performance, with accuracy at 0.94, precision at 0.81, recall at 0.92, and an F1-score of 0.86 for fraud detection, all with minimal computational costs (training time of 0.09 s and testing time of 0.00 s). Conversely, LOF and SVM show significantly lower performance, particularly in fraud detection, with LOF and SVM achieving F1-scores of 0.14 and 0.35, respectively. RNN and IF offer moderate performance; however, RNN's high training time of 8.88 s is a notable drawback.

These results have significant implications for the Ethereum classification problem. Given the high volume and complexity of transactions on the Ethereum network, the ability to accurately and efficiently detect fraudulent activities is crucial. The superior performance of RF and XGBoost indicates that these models can effectively identify fraudulent transactions with high precision and recall, minimizing false positives and false negatives. This ensures that legitimate transactions are not wrongly flagged, maintaining the integrity of the network. Furthermore, the efficiency of these models in terms of computational cost makes them suitable for real-time fraud detection, essential for the dynamic and fast-paced environment of blockchain transactions. Logistic regression, with its balance of performance and computational efficiency, offers a practical alternative for scenarios with limited computational resources. In contrast, the lower performance of LOF and SVM suggests that these models may not be adequate for the Ethereum classification problem without significant tuning. The moderate performance of RNN and IF, along with RNN's high training time, indicates that while these models have potential, they may not be the most practical choices given the need for timely detection. Overall, the findings
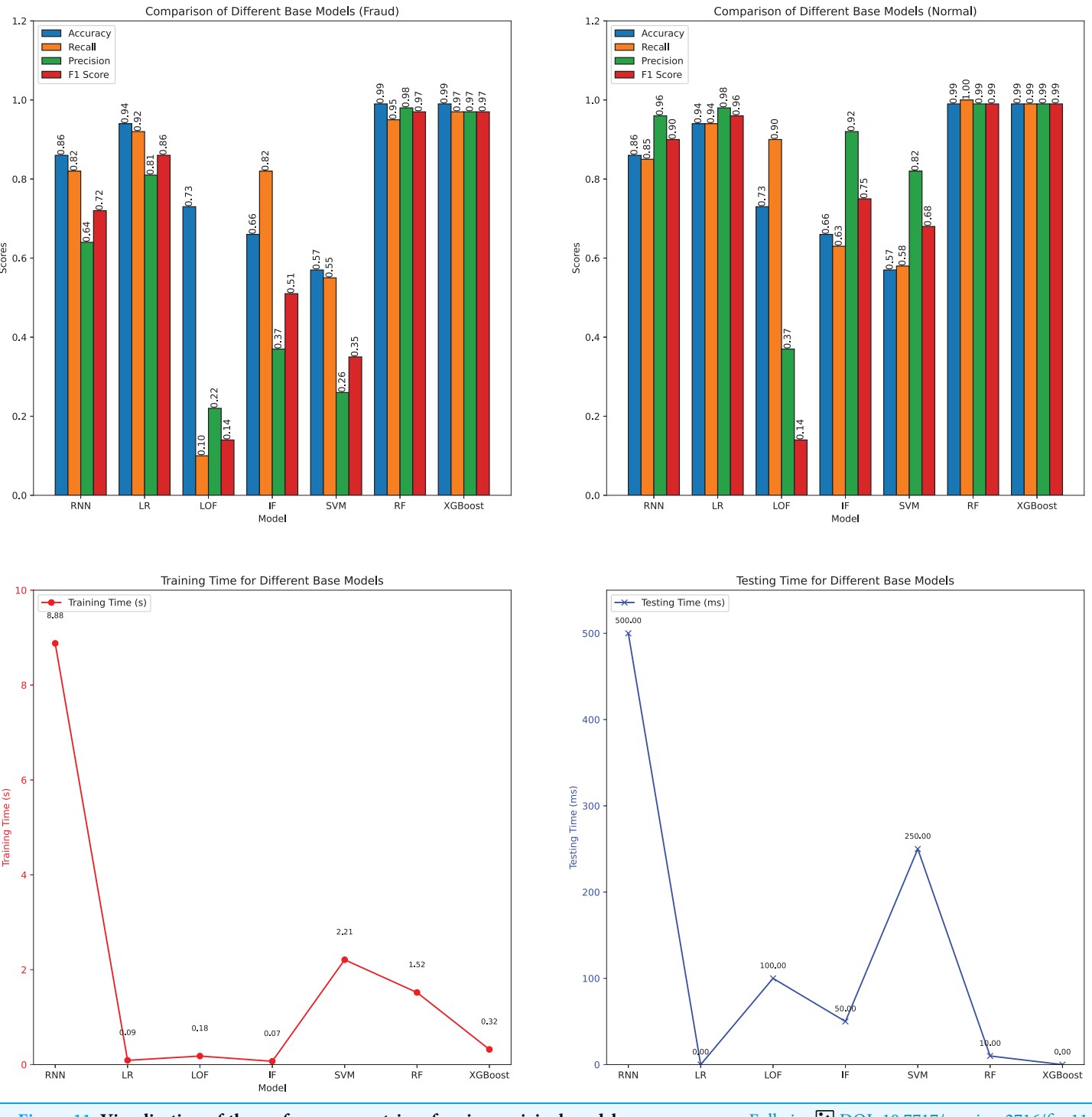

**Figure 11 Visualization of the performance metrics of various original models.**

**Table 2 Performance metrics of different grided models.**

| Model | Instance | Accuracy | Precision | Recall | F1-score | Training (s) | Testing (s) |
|---|---|---|---|---|---|---|---|
| RNN | Fraud | 0.79 | 0.70 | 0.07 | 0.13 | 40.17 | 0.45 |
| | Normal | 0.79 | 0.80 | 0.99 | 0.88 | | |
| LR | Fraud | 0.94 | 0.81 | 0.94 | 0.96 | 1.13 | 0.00 |
| | Normal | 0.94 | 0.98 | 0.94 | 0.96 | | |
| LOF | Fraud | 0.77 | 0.37 | 0.09 | 0.14 | 8.50 | 0.19 |
| | Normal | 0.77 | 0.79 | 0.96 | 0.87 | | |
| IF | Fraud | 0.28 | 0.23 | 0.98 | 0.37 | 41.67 | 0.05 |
| | Normal | 0.28 | 0.93 | 0.08 | 0.15 | | |
| SVM | Fraud | 0.98 | 0.92 | 0.97 | 0.94 | 77.83 | 0.08 |
| | Normal | 0.98 | 0.99 | 0.98 | 0.98 | | |
| RF | Fraud | 0.99 | 0.98 | 0.95 | 0.97 | 1,572.28 | 0.03 |
| | Normal | 0.99 | 0.99 | 0.99 | 0.99 | | |
| XGBoost | Fraud | 0.99 | 0.96 | 0.98 | 0.97 | 408.53 | 0.00 |
| | Normal | 0.99 | 0.99 | 0.99 | 0.99 | | |

highlight RF and XGBoost as the most promising models for enhancing fraud detection on the Ethereum network, ensuring both accuracy and efficiency.

The parameters of each model were fine-tuned using Grid Search technology, which systematically traverses a predefined parameter grid to evaluate the performance of each parameter combination. This process identifies the optimal set of parameters to enhance model performance. During fine-tuning, parameters such as learning rate, tree depth, and regularization factors are adjusted to maximize performance on the training set and ensure robust generalization. The performance metrics of the fine-tuned versions of various classification algorithms are presented in Table 2 and further illustrated in Fig. 12.

After fine-tuning the parameters using Grid Search, the performance metrics of the models show notable improvements, especially in the classification of fraud instances. RF and XGBoost continue to be the top performers, achieving an accuracy of 0.99 for both fraud and normal instances. RF's precision and recall for fraud detection are 0.98 and 0.95, respectively, while XGBoost achieves 0.96 and 0.98. These results confirm the robustness and efficacy of ensemble methods in handling the complexity of Ethereum transactions.

The SVM model shows significant improvement after fine-tuning, with its accuracy for fraud detection increasing to 0.98, precision at 0.92, recall at 0.97, and F1-score at 0.94. This enhancement underscores the importance of parameter optimization in boosting the model's performance for fraud detection. LR also benefits from fine-tuning, achieving an accuracy of 0.94 for both fraud and normal instances, with an F1-score of 0.96 for fraud detection, indicating balanced performance across both categories.

However, not all models showed substantial improvements. The RNN demonstrated a decrease in performance for fraud detection, with accuracy dropping to 0.79 and an F1-score of 0.13, despite achieving an F1-score of 0.88 for normal instances. This suggests that

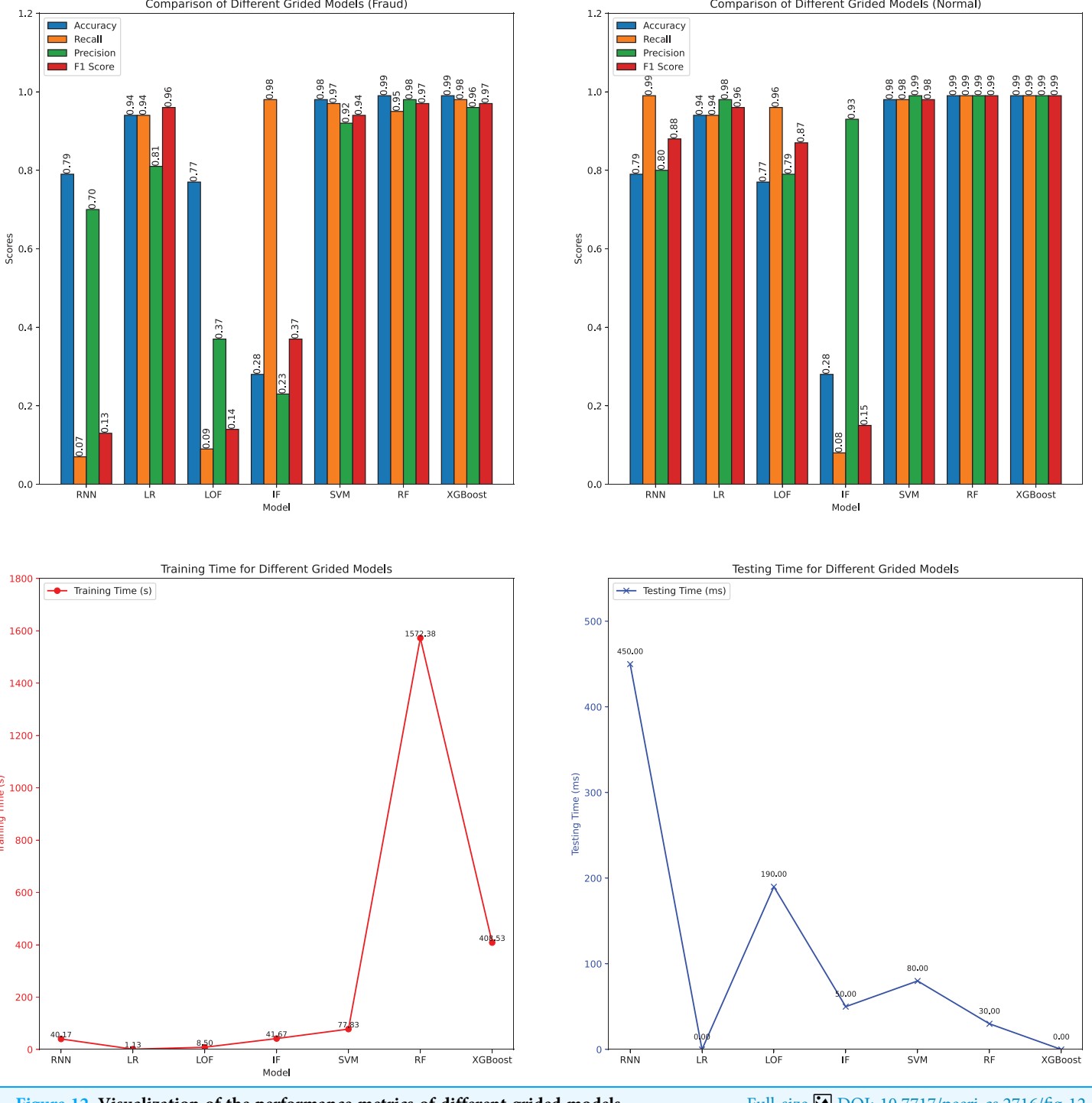

**Figure 12** Visualization of the performance metrics of different grided models.

while RNNs are effective for normal transaction detection, they may struggle with the nuances of fraud detection even after parameter optimization.

The LOF model also showed limited improvement, with accuracy and F1-score for fraud detection at 0.77 and 0.14, respectively. Similarly, the Isolation Forest (IF) model performed poorly for normal instances, with an accuracy of 0.28 and an F1-score of 0.15, despite achieving a high recall of 0.98 for fraud detection. These results highlight the limitations of anomaly detection methods in this context.

The training times of the models increased significantly after applying Grid Search, reflecting the computational cost of parameter fine-tuning. RF and XGBoost training times rose to 1,572.28 s and 408.53 s, respectively, which is a substantial increase compared to their initial training times. This trade-off between enhanced performance and increased computational expense is an important consideration for real-time fraud detection in the Ethereum network.

To further enhance the robustness of the fraud classification framework, we have chosen RF, SVM, and XGBoost for integration into three distinct ensemble learning methods: voting, stacking, and boosting. These individual models were selected as base models due to their diverse performance characteristics and unique properties, which are advantageous for ensemble techniques. In ensemble learning, base models should ideally be as varied as possible to enhance performance.

After fine-tuning through Grid Search, the optimal parameters for each model are as follows:

- Support Vector Machine: $C = 10$, $gamma = scale$, $kernel = rbf$.
- Random Forest: $max\_depth = None$, $max\_features = auto$, $min\_samples\_leaf = 1$, $min\_samples\_split = 2$, $n\_estimators = 100$.
- XGBoost: $colsample\_bytree = 0.7$, $learning\_rate = 0.5$, $max\_depth = 4$, $n\_estimators = 200$, $subsample = 0.9$.

Ensemble learning models can be trained more effectively by integrating these well-performing base models. In the Boosting method, errors from each model's training are iteratively passed to the next model using the gradient boosting algorithm, resulting in a highly performant Boosting model. For the stacking and voting methods, the Optuna technique was employed to select optimal parameters, further enhancing model performance.

The performance metrics of the three ensemble methods—voting, stacking, and boosting—are presented in Table 3 and further illustrated in Fig. 13.

The results show that for the fraud detection category, both voting and stacking achieved high performance metrics, with accuracy, precision, recall, and F1-score, all reaching 0.99. Boosting, while slightly behind in recall (0.95), still maintained a strong overall performance with an accuracy of 0.98 and F1-score of 0.96. This suggests that voting and stacking effectively leverage the combined strengths of their constituent models to achieve robust fraud detection, whereas Boosting focuses on iterative error correction to enhance predictive accuracy.

**Table 3 Performance metrics of different ensemble models.**

| Model | Instance | Accuracy | Precision | Recall | F1-score | Training (s) | Testing (s) |
|---|---|---|---|---|---|---|---|
| Voting | Fraud | 0.99 | 0.98 | 0.97 | 0.98 | 134.00 | 0.13 |
| | Normal | 0.99 | 0.99 | 0.99 | 0.99 | | |
| Stacking | Fraud | 0.99 | 0.98 | 0.97 | 0.97 | 987.09 | 0.37 |
| | Normal | 0.99 | 0.99 | 0.99 | 0.99 | | |
| Boosting | Fraud | 0.98 | 0.98 | 0.95 | 0.96 | 3.99 | 0.02 |
| | Normal | 0.98 | 0.99 | 0.99 | 0.99 | | |

In contrast, across the normal category, all three ensemble methods performed uniformly well, each achieving an accuracy, precision, recall, and F1-score of 0.99. This consistency indicates that ensemble methods effectively generalize across normal transactions, maintaining high performance levels without significant variation.

Regarding computational efficiency, Boosting emerged as the fastest in both training and testing times, with training completed in 3.99 s and testing in 0.02 s. Stacking, on the other hand, exhibited the longest training time at 987.09 s and a testing time of 0.37 s. Voting occupied an intermediate position in terms of computational efficiency.

The application of ensemble learning methods has notably enhanced the performance of individual models. By combining the predictions of diverse base models, ensemble methods not only improve accuracy but also enhance robustness against fraudulent transactions. Voting emphasizes overall model stability and performance consistency, while boosting excels in refining predictive accuracy through iterative learning. Stacking offers a more nuanced approach, allowing for complex model interactions and deeper performance analysis, albeit at the cost of increased computational time.

## Comparative analysis

The results presented in Table 4 provide a comprehensive comparative analysis of the best results from five referenced studies alongside our project results. Each study, including ours, utilized the same dataset to ensure consistency in comparison. The performance metrics reported for the five referenced articles reflect their results on the entire test set, whereas our project specifically details the performance on both Fraud and Normal labels individually.

The referenced works include:

- *Aziz et al. (2022a)*, which applied LightGBM (LGBM) and optimized its hyperparameters using Euclidean distance as the loss function with random search.
- *Aziz et al. (2022b)*, which also used LGBM and optimized its hyperparameters with random search.
- *Aziz et al. (2023)*, which employed a deep learning artificial neural network (DLANN) and optimized its hyperparameters using a genetic algorithm and cuckoo search.
- *Md et al. (2023)*, which utilized a stacking classifier model with multinomial naive Bayes and RF as base learners, and logistic regression as the meta-learner.

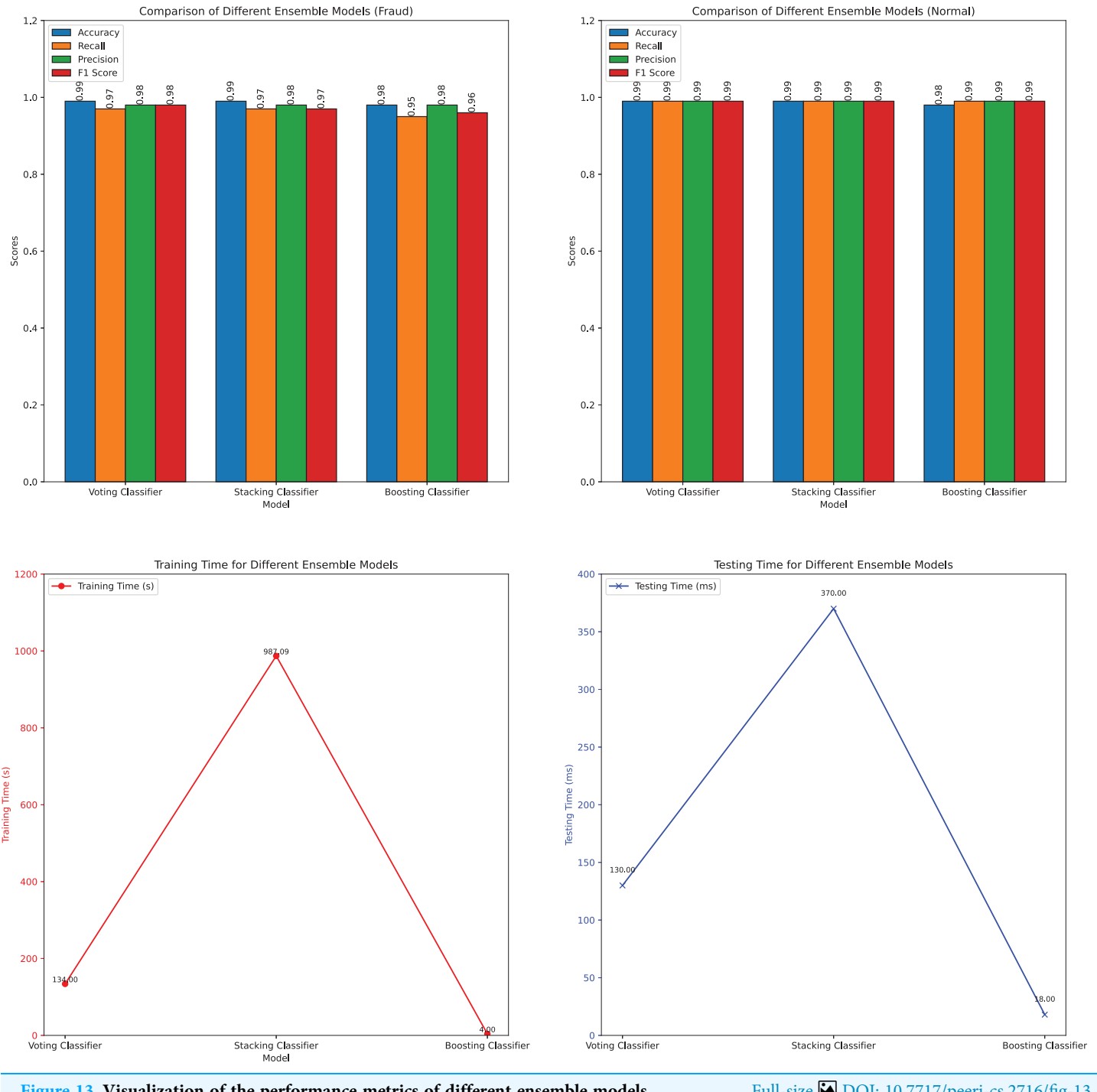

**Figure 13** Visualization of the performance metrics of different ensemble models.

- *Ravindranath et al. (2024)*, which conducted an extensive evaluation of fraud detection methods by implementing ten machine learning models under various preprocessing settings, including without oversampling, with SMOTENC, and with K-Means-SMOTE. Notably, the study employed an XGBoost model optimized using K-Means-SMOTE for

**Table 4 Benchmarking against existing fraud detection models.**

| References | Acc | Prec | Recall | F1 | Training (s) | Testing (s) | Code |
|---|---|---|---|---|---|---|---|
| *Aziz et al. (2022a)* | 0.99 | 0.97 | 0.93 | 0.95 | No | No | No |
| *Aziz et al. (2022b)* | 0.99 | 0.99 | 0.99 | 0.99 | No | No | No |
| *Aziz et al. (2023)* | 0.99 | 0.96 | 0.94 | 0.97 | No | No | No |
| *Md et al. (2023)* | 0.97 | 0.97 | 0.97 | 0.97 | No | No | No |
| *Ravindranath et al. (2024)* | 0.98 | 0.98 | 0.97 | 0.97 | No | 0.235 | No |
| Our Framework | 0.99 | 0.98 | 0.97 | 0.98 | 134.00 | 0.13 | Yes |
| | 0.99 | 0.99 | 0.99 | 0.99 | | | |

oversampling, demonstrating the effectiveness of advanced oversampling techniques in enhancing model performance. This comprehensive approach provides valuable benchmarks for comparison and highlights the impact of different preprocessing strategies on fraud detection accuracy.

This comparative analysis highlights the varied approaches and optimization techniques applied across different studies, providing a contextual understanding of our project's performance in relation to existing methodologies.

Results in Table 4 indicate that the referenced studies exhibit high accuracy (Acc), precision (Prec), recall, and F1 scores, demonstrating their robust performance on the dataset. Notably, *Aziz et al. (2022b)* achieved exceptional performance, with all metrics reaching 0.99. Other studies, such as *Aziz et al. (2022a)* and *Aziz et al. (2023)*, also show strong results with slight variations in precision, recall, and F1-scores. However, none of these studies report the training time, and only *Ravindranath et al. (2024)* reports the execution time. Moreover, the source code for all these studies is not available, making it difficult to reproduce the reported results.

In contrast to those approaches, our project employed several data pre-processing techniques, hyper-parameter optimization tools, and ensemble learning methods such as the voting classifier, which integrates multiple models to enhance overall performance. The results indicate that our best mode, the voting classifier, achieves an accuracy of 0.99, with precision, recall, and F1-scores consistently high across both labels (0.98 and 0.99). The training and testing times for our model were 134.00 s and 0.13 s, respectively, showcasing the efficiency of our approach. This comprehensive strategy not only matches but often surpasses the performance metrics of the referenced studies, highlighting the robustness and effectiveness of our methodology. This performance is comparable to the best results from the referenced studies, but our method also offers transparency and reproducibility, as the source code for our implementation is available. This ensures that our results can be independently verified and applied in different contexts, enhancing the robustness and reliability of our findings. Additionally, our comprehensive reporting of both training and testing times provides a clearer understanding of the computational efficiency of our method, an aspect that is often overlooked in other studies. Besides, a key advantage of our model lies in its balanced and superior performance metrics across both Fraud and Normal

labels. This consistency is crucial for applications where distinguishing between these labels with high precision and recall is necessary. Moreover, the efficiency of our voting classifier is highlighted by its quick training and testing times, making it a practical and robust solution for real-world applications where time and computational resources are limited.

The comparative analysis, therefore, highlights that while the referenced studies provide strong baseline performances, our best model, the voting classifier, matches these benchmarks and surpasses them in terms of precision, recall, and F1-score. Additionally, our model demonstrates competitive computational efficiency, with training and testing times of 134.00 s and 0.13 s respectively, offering a significant advantage in practical deployment scenarios. This efficiency ensures that our approach is both scalable and practical for real-world applications, particularly in environments where rapid and accurate fraud detection is critical.

## CONCLUSION

This research investigates the application of ensemble learning methods to enhance the detection of fraudulent transactions within the Ethereum blockchain. The exponential growth of online commerce has made the Ethereum platform a prime target for fraudulent activities such as money laundering and phishing, exacerbating security vulnerabilities. To address these challenges, this study introduces the Ensemble Stacking Machine Learning (ESML) approach for accurately detecting fraudulent transactions.

The methodology employed involved rigorous data preprocessing and the evaluation of multiple machine learning algorithms, including LR, IF, SVM, RF, XGBoost, and RNN. Each model was fine-tuned using grid search to optimize performance metrics. The models were then integrated into ensemble learning methods: voting, stacking, and boosting, to leverage their diverse performance characteristics and enhance overall classification performance.

The experimental results demonstrated that ensemble learning methods significantly improve the robustness and accuracy of fraud detection models. Both voting and stacking methods achieved high performance metrics, with accuracies, precision, recall, and F1-scores all reaching 0.99 for both fraudulent and normal transactions. Boosting, while slightly lower in recall for fraud detection at 0.95, maintained high overall performance, underscoring its effectiveness in iterative error correction.

The impact of this research is substantial, as it not only enhances the security of the Ethereum blockchain but also provides a robust framework that can be integrated into the decentralized validation process. This integration allows miners to identify and flag fraudulent transactions effectively, and assists regulatory bodies in monitoring and mitigating fraudulent activities. The ensemble methods, especially stacking, which achieved high scores across all key metrics with an inference time of 0.37 s, are suitable for real-world applications despite their computational intensity.

Future work will further optimize these models to reduce training and inference times, making them more suitable for real-time applications in high-frequency trading environments. Additionally, exploring other machine learning and deep learning

techniques, such as reinforcement learning and neural architecture search, could further enhance the performance and robustness of fraud detection systems.

In addition to addressing data limitations, future work will focus on integrating external data sources to improve the model's robustness and generalizability. This could include incorporating real-time transaction data from the Ethereum blockchain or external threat intelligence feeds to enrich the training process and provide a broader context for fraud detection. By expanding the scope of the dataset and considering additional data points, we aim to reduce the impact of biases and better capture the complexities of fraudulent activities. Moreover, this approach could enhance the model's ability to adapt to new fraud patterns, further improving its effectiveness in dynamic and rapidly changing environments.

## ACKNOWLEDGEMENTS

The authors would like to express their sincere gratitude to the reviewers for their thorough and insightful comments. Their constructive feedback has been invaluable in refining our research and enhancing the overall quality of this article.

### Funding

This research was funded by the Wenzhou-Kean University Computer Science and Artificial Intelligence Center (Project No. BM20211203000113), Wenzhou-Kean University Student Partnering with Faculty (Project No. WKUSPF202444), Wenzhou-Kean University Internal (Faculty/Staff) Research Support Program (IRSP) (Project No. IRSPG202105), and Wenzhou-Kean University International Collaborative Research Program (Project No. ICRP2023008). The funders had no role in study design, data collection and analysis, decision to publish, or preparation of the manuscript.

### Grant Disclosures

The following grant information was disclosed by the authors:
Wenzhou-Kean University Computer Science and Artificial Intelligence Center: BM20211203000113.
Wenzhou-Kean University Student Partnering with Faculty: WKUSPF202444.
Wenzhou-Kean University Internal (Faculty/Staff) Research Support Program (IRSP): IRSPG202105.
Wenzhou-Kean University International Collaborative Research Program: ICRP2023008.

### Competing Interests

The authors declare that they have no competing interests.

### Author Contributions

- Zhexian Gu conceived and designed the experiments, performed the experiments, analyzed the data, performed the computation work, prepared figures and/or tables, authored or reviewed drafts of the article, and approved the final draft.

- Omar Dib conceived and designed the experiments, performed the experiments, analyzed the data, performed the computation work, prepared figures and/or tables, authored or reviewed drafts of the article, and approved the final draft.

## Data Availability

The models are available at GitHub and Zenodo:

- https://github.com/Lemoninmountain/Enhancing-Fraud-Detection-in-the-Ethereum-Blockchain-Using-Ensemble-Stacking-Machine-Learning.git

- Gary_Guzh. (2025). Lemoninmountain/Enhancing-Fraud-Detection-in-the-Ethereum-Blockchain-Using-Ensemble-Stacking-Machine-Learning: v1.0.0 (v1.0.0). Zenodo. https://doi.org/10.5281/zenodo.14718520

## Supplemental Information

Supplemental information for this article can be found online at http://dx.doi.org/10.7717/peerj-cs.2716#supplemental-information.

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
