# Peer review of "Enhancing fraud detection in the Ethereum blockchain using ensemble learning"

_PeerJ Computer Science, doi:10.7717/peerj-cs.2716_

## Round 0.1 · original submission · Major Revisions

Dear authors,

Thanks a lot for your manuscript.

Please address the issues relating to model selection justification and claims of real-world applicability. Additionally, the comparison with existing methods is limited while some sections are overly detailed, affecting readability. A streamlined presentation and clearer results would enhance the paper’s quality.

Kind regards,
PCoelho

Reviewer 1 ·

Basic reporting

This work proposes an Ethereum blockchain enhancement through an ensemble approach to detect fraud upon generating the transaction stored in the blockchain.

Clear and unambiguously written. Although it is a lengthy article.

Found a couple of typos (eg., 173 - Random, 603 - exponential, 605 - these, 609 - was, 610 - were, 614 - and, 616 - while, 620 - integration )

Introduction and Related Work are relevant and demonstrate context. References are well-referenced and are relevant despite being easier to read with another bibliographical notation (IEEE Numerical Reference System).

Experimental design

The article falls within the context of the journal and provides enough information in its main text and also in supplementary materials that promote transparency and enough information to replicate the experiment.

The method is fully described and each stage is documented.

The amount of references is adequate but the authors present a blockchain introduction, specifically to Bitcoin and Ethereum, thus, consider the following 2 references:
- Nakamoto, S. (2008). Bitcoin: A peer-to-peer electronic cash system.
- Buterin, V. (2013). Ethereum white paper. GitHub repository, 1, 22-23.

Validity of the findings

No comment.

Additional comments

This work needs only a final proofreading.

It is a detailed work with full supplementary materials that allow researchers to completely replicate the experiment.

I congratulate the authors on the work accomplished and the best for the upcoming steps in this line of work.

Cite this review as

Reviewer 2 ·

Basic reporting

Paper is well structured.

Experimental design

Experimetal results are acceptable.

Validity of the findings

Good

Additional comments

NA

Cite this review as

·

Basic reporting

The paper is valuable but requires significant revisions for clarity, real-world validation, and a stronger comparison with existing methods.

The paper uses clear and professional language. However, some sections, particularly the introduction and results, are overly detailed. These could be streamlined for conciseness.

The background and literature review are comprehensive, but the paper could benefit from a clearer focus on how it builds upon past work.

The overall structure conforms to PeerJ standards, though better transitions between sections would improve flow.

Experimental design

The methodology is well-described, with sufficient detail for reproducibility. However, more explanation is needed for the model selection process, especially why specific ensemble models were chosen.

Data preprocessing steps are thorough, but more information is required on handling imbalanced data, especially with respect to fraudulent transactions.

While the evaluation metrics are clear, the reasoning behind choosing certain metrics should be elaborated further.

Validity of the findings

The results are promising, particularly the accuracy of over 98%. However, the paper should present more comparison with state-of-the-art models in fraud detection on blockchain platforms.

The paper claims real-world usage, but this could be substantiated further with examples of how the system might operate in practice.

Additional comments

The figures are well-presented but could benefit from more descriptive captions, explaining their relevance to the results.

The dataset source is well-cited, but additional discussion on data limitations (e.g., bias or missing data) is needed.

---

## Round 0.2 · accepted · Accept

Dear authors, we are pleased to verify that you meet the reviewer's valuable feedback to improve your research.

Thank you for considering PeerJ Computer Science and submitting your work.

Kind regards
PCoelho

Reviewer 1 ·

Basic reporting

no comment

Experimental design

no comment

Validity of the findings

no comment

Additional comments

Figure 5 could be easily resized to take less space without losing its intent.

Cite this review as

·

Basic reporting

The authors have addressed my comments

Experimental design

The authors have addressed my comments

Validity of the findings

The authors have addressed my comments

Additional comments

The authors have addressed my comments